# Last-Mile Embodied Visual Navigation

**Justin Wasserman**[1*†], **Karmesh Yadav**[2], **Girish Chowdhary**[1†],
**Abhinav Gupta**[3], **Unnat Jain**[2*]
[1]University of Illinois at Urbana-Champaign,
[2]Meta AI Research, [3]Carnegie Mellon University

**Abstract:** Realistic long-horizon tasks like image-goal navigation involve exploratory and exploitative phases. Assigned with an image of the goal, an embodied agent must explore to *discover the goal*, *i.e.*, search efficiently using learned priors. Once the goal is discovered, the agent must accurately calibrate the *last-mile of navigation* to the goal. As with any robust system, *switches* between exploratory goal discovery and exploitative last-mile navigation enable better recovery from errors. Following these intuitive guide rails, we propose SLING to improve the performance of existing image-goal navigation systems. Entirely complementing prior methods, we focus on last-mile navigation and leverage the underlying geometric structure of the problem with neural descriptors. With simple but effective switches, we can easily connect SLING with heuristic, reinforcement learning, and neural modular policies. On a standardized image-goal navigation benchmark [1], we improve performance across policies, scenes, and episode complexity, raising the state-of-the-art from 45% to 55% success rate. Beyond photorealistic simulation, we conduct real-robot experiments in three physical scenes and find these improvements to transfer well to real environments. Code and results: https://jbwasse2.github.io/portfolio/SLING

**Keywords:** Embodied AI, Robot Learning, Visual Navigation, Perspective-n-Point, AI Habitat, Sim-to-Real.

## 1 Introduction

Imagine you are at a friend's home and you want to find the couch you have seen in your friend's photo. At first, you use semantic priors *i.e.* priors about the semantic structure of the world, to navigate to the living room (a likely place for the couch). But as soon as you get the first glimpse of the couch, you implicitly estimate the relative position of the couch, use intuitive geometry, and navigate towards it. We term the latter problem, of navigating to a visible object or region, as last-mile navigation.

The field of visual navigation has a rich history. Early approaches used hand-designed features with geometry for mapping followed by standard planning algorithms. But such an approach fails to capture the necessary semantic priors that could be learned from data. Therefore, in recent years, we have seen more efforts and significant advances in capturing these priors for semantic navigation tasks such as image-goal [2, 3, 4, 1, 5, 6] and object-goal navigation [7, 8, 9, 10]. The core idea is to train a navigation policy using reinforcement or imitation learning and capture semantics. But in an effort to capture the semantic priors, these approaches almost entirely bypass the underlying geometric structure of the problem, specifically when the object or view of interest has already been discovered.

One can argue that last-mile navigation can indeed be learned from data itself. We agree that, in principle, it can be. However, we argue and demonstrate that an unstructured local policy for last-mile navigation is either (a) sample inefficient (billions of frames in an RL framework [11]) or (b) biased and generalize poorly when learned from offline demonstrations (due to distributional shift [12, 13]). Therefore, our solution is to revisit the basics! We propose **S**witchable **L**ast-Mile

---

*equal technical contribution; †corresponding authors

6th Conference on Robot Learning (CoRL 2022), Auckland, New Zealand.

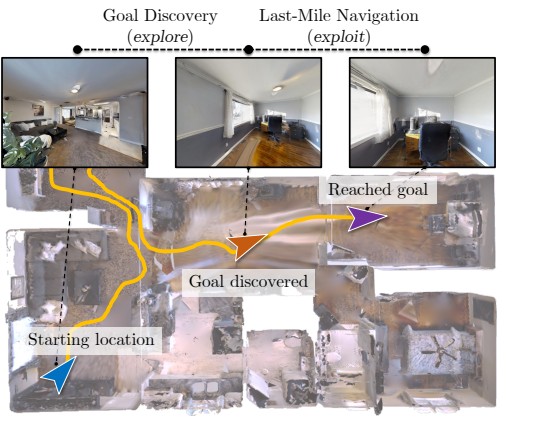
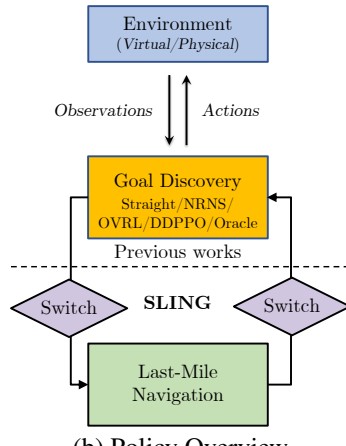

|  |  |
|---|---|
| (a) Phases in Image-Goal Navigation | (b) Policy Overview |

Figure 1: **Switchable Last-Mile Image-Goal Navigation**. (a) Long-horizon semantic tasks such as image-goal navigation involves exploratory discovery of goals and exploitative last-mile navigation, (b) An overview of SLING that allows for *switching* between policies from prior work and our last-mile navigation system.

**I**mage-Goal **N**avi**g**ation (SLING) – a simple yet very effective geometric navigation system and associated switches. Our approach can be combined with any off-the-shelf learned policy that uses semantic priors to explore the scene. As soon as the object or view of interest is detected, the SLING switches to the geometric navigation system. We observe that SLING provides significant performance gains across baselines, simulation datasets, episode difficulty, and real-world scenes.

Our key contributions are: (1) A general-purpose last-mile navigation system and switches, that we connect with five diverse goal discovery methods, leading to improvements across the board. (2) A new state-of-the-art of 54.8% success *i.e.* a huge jump of 21.8% *vs.* published work [5] and 9.2% *vs.* a concurrent preprint [6], on the most widely-tested fold (Gibson-curved) of the AI Habitat image-goal navigation benchmark [1]; (3) Extensive robot experiments of image-goal navigation in challenging settings with improved performance over a neural, modular policy [1] trained on real-world data [14].

## 2 Related Work

Prior work in visual navigation and geometric 3D vision is pertinent to SLING.

**Embodied navigation.** Anderson *et al*. [15] formalized different goal definitions and metrics for the evaluation of embodied agents. In point-goal navigation, relative coordinates of the goal are available (either at all steps [16, 11, 17, 18, 19] or just at the start of an episode [9, 20, 21]). Successful navigation to a point-goal could be done without semantic scene understanding, as seen by competitive depth-only agents [16, 11]. Semantic navigation entails identifying the goal through an image (image-goal [1, 2, 22]), acoustic cues (audio-goal [23, 24]), or a category label (object-goal [8, 9]). Several extensions of navigation include language-conditioned navigation following [25, 26, 27, 28], social navigation [29, 30, 31, 32, 33], and multi-agent tasks [34, 35, 36, 37, 38, 39]. However, each of these build-off single-agent navigation and benefit from associated advancements. For more embodied tasks and paradigms, we refer the reader to a recent survey [40]. In this work, we focus on image-goal navigation in visually rich environments.

**Image-goal navigation.** Chaplot *et al*. [3] introduced a modular and hierarchical method for navigating to an image-goal that utilizes a topological map memory. Kwon *et al*. [41] introduced a memory representation based on image similarity, which in turn is learned in an unsupervised fashion from unlabeled data and the agent's observed images. Following up on [3], NRNS [1] improves the topological-graph-based architecture and open-sourced a public dataset and IL and RL baselines [11, 3] within AI Habitat. This dataset has been adopted for standardized evaluation [5, 6]. ZER [5] focuses on transferring an image-goal navigation policy to other navigation tasks. In a concurrent preprint, Yadav *et al*. [6] utilize self-supervised pretraining [42] to improve an end-to-end visual RL policy [11] for the image-goal navigation benchmark. Our contributions are orthogonal to the above and can be easily combined with them, as we demonstrate in Sec. 4.

Beyond simulation, SLING finds relevance to the rich literature of navigating to an image-goal on physical robots. Meng *et al.* [4] utilize a neural reachability estimator and a local controller based on a Riemannian Motion Policy framework to navigate to image-goals. Hirose *et al.* [43] train a deep model predictive control policy to follow a trajectory given by a sequence of images while being robust to variations in the environment. Even in outdoor settings, meticulous studies have shown great promise, based on negative mining, graph pruning, and waypoint prediction [44] and utilizing geographic hints for kilometer-long navigations [45]. Complementing this body of work, SLING tackles image-goal navigation in challenging indoor settings, without needing any prior data of the test environment (similar to [1, 6, 3]) *i.e.* during evaluation no access to information (trajectories, GPS, or top-down maps) in the test scenes is assumed.

**Last-mile navigation.** The works included above focus primarily on goal discovery. In contrast, recent works have also identified 'last-mile' errors that occur when the goal is in sight of or close to the agent. For multi-object navigation, Wani *et al.* [46, 47] observed a two-fold improvement when allowing an error budget for the final 'found' or 'stop' actions. Chattopadhyay *et al.* [48] found the last step of navigation to be brittle *i.e.* small perturbations lead to severe failures. Ye *et al.* [10] identified last-mile errors as a prominent error mode (10% of the failures) in object-goal navigation. However, none of these works address the problem with the last-mile of navigation. From a study inspired by [46], we infer that better (or more tolerant to error) last-mile navigation can indeed lead to better performance in the image-goal navigation task (details in Appendix H).

**Connections to 3D vision.** The objective of our last-mile navigation system is to predict the relative camera pose between two images *i.e.* agent's view and image-goal. To this end, pose estimation of a calibrated camera from 3D-to-2D point correspondences connects our embodied navigation task to geometric 3D computer vision. The Perspective-n-Point (PnP) formulation, with extensive research and efficient solvers [49, 50, 51], fits this use case perfectly. To find an accurate PnP solution, locating correspondences between the local features of the two images is critical. We utilize SuperGlue [52] which is based on correspondences learned via attention graph neural nets and partial assignments. We defer details of PnP and finding correspondences to Sec. 3.3, to make the approach self-sufficient. Notably, different from related works in 3D vision [53, 54, 55], we apply SLING to sequential decision-making in embodied settings, particularly, image-goal navigation. To take policies to the real world, we utilized robust SLAM methods [56, 57] for local odometry and pose estimation, which has also been found reliable by prior works in sim-to-real [58, 59, 60].

## 3 SLING

In this section, we begin with an overview of the task and the entire pipeline of SLING. We then discuss the implementations for goal discovery, our proposed system for last-mile navigation, and switches to easily combine it with prior works. While we explain key design choices in the main paper, a supplementary description and a list of hyperparameters, for effective reproducibility, is deferred to Appendix A.

### 3.1 Overview

We follow the image-goal navigation task benchmark by Hahn *et al.* [1] (similar to the prior formulations [2, 3]). The agent observes an RGB image $\mathbf{I}_a$, a depth map $\mathbf{D}_a$, and the image-goal $\mathbf{I}_g$. The agent can sample actions from $\mathcal{A} = \{$move forward, turn right, turn left, stop$\}$. The stop action terminates the episode.

As shown in Fig. 1a, we divide image-goal navigation into – a goal discovery and a last-mile navigation phase. In the goal discovery phase, the agent is responsible for discovering the goal *i.e.* navigating close enough for the goal to occupy a large portion of the egocentric observation ('goal discovered' image). Fig. 1b shows how the control flows between our system. If the *explore→exploit* switch isn't triggered, learning-based exploration will continue. Otherwise, if the *explore→exploit* switch triggers, the agent's observations now overlap with the image-goal and the control flows to the last-mile navigation system. We find that a one-sided flow (as attempted in [1, 3]) from *explore→exploit* is too optimistic. Therefore, we introduce symmetric switches, including one that flows control back to goal discovery.

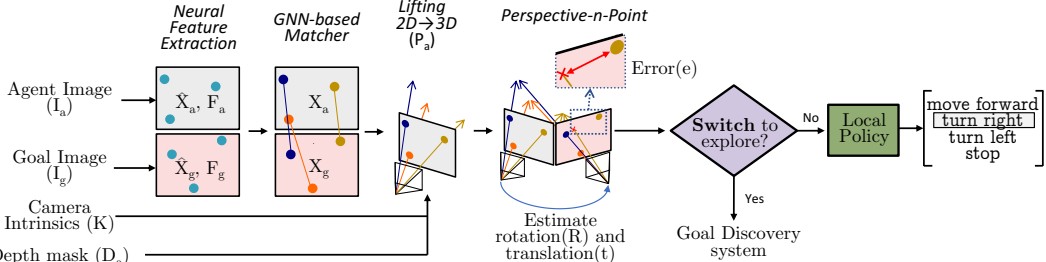

Figure 2: **Last-Mile Navigation system.** Neural keypoint feature descriptors are extracted and matched to obtain correspondences between the agent's view and the image-goal. The geometric problem of estimating the relative pose between the agent and goal view is solved using efficient perspective-n-point. A *exploit→explore* switch, if triggered, flows control back to the goal discovery phase. Else, the estimations are fed into a local policy head to decide the agent's actions.

## 3.2   Goal Discovery

We can combine our versatile last-mile navigation system and switching mechanism with any prior method. These prior methods are previously suggested solutions to image-goal navigation. We demonstrate this with five diverse goal discovery (GD) implementations.

**Straight [61].** A simple, heuristic exploration where the agent moves forward and unblocks itself, if stuck, by turning right (similar to an effective exploration baseline in [61]).

**Distance Prediction Network (NRNS-GD) [1].** Exploratory navigation is done by proposing way-points in navigable areas (determined utilizing the agent's depth mask), history is maintained using a topological map, and processed using graph neural nets. The minimum cost waypoint is chosen utilizing outputs from a distance prediction network. More details are given in Appendix B and [1].

**Decentralized Distributed PPO (DDPPO-GD) [11].** An implementation of PPO [62] for photorealistic simulators where rendering is the computational bottleneck. This has been a standard end-to-end deep RL baseline in prior works, across tasks [18, 1, 5, 6, 63].

**Offline Visual Representation Learning (OVRL-GD) [6].** A DDPPO network, with its visual encoder pretrained using self-supervised pretext tasks [42] on images obtained from 3D scans [64].

**Environment-State Distance Prediction (Oracle-GD).** To quantify the effect of errors coming from the goal discovery phase, we devise an upper bound. This is a privileged variant of NRNS-GD that accesses the ground-truth distances from the environment, exclusively for the goal discovery phase. For fine details of its construction, particularly, how we curtail this to be an oracle explorer and not an oracle policy, see Appendix B.

## 3.3   Last-Mile Navigation

The proposed last-mile navigation module transforms the agent's observations and image-goal into actions that take the agent closer to the goal. The steps are shown in Fig. 2 and detailed next.

**Neural Feature Extractor.** We first transform the agent's RGB $\mathbf{I}_a$ to local features $(\hat{\mathbf{X}}_{\mathbf{a}}, \mathbf{F}_a)$, where $\hat{\mathbf{X}}_{\mathbf{a}} \in \mathbb{R}^{n_a \times 2}$ are the positions and $\mathbf{F}_a \in \mathbb{R}^{n_a \times k}$ are the visual descriptors in the agent's image. Here, $n_a$ is the number of detected local features and $k$ is the length of each descriptor. Similarly, $\mathbf{I}_g$ leads to features $(\hat{\mathbf{X}}_{\mathbf{g}}, \mathbf{F}_g)$, where $\hat{\mathbf{X}}_{\mathbf{g}} \in \mathbb{R}^{n_g \times 2}$ and $\mathbf{F}_g \in \mathbb{R}^{n_g \times k}$ with $n_g$ local features in the image-goal. Following DeTone *et al.* [65], we adopt an interest-point detector, pretrained on synthetic data followed by cross-domain homography adaptation (here, $k = 256$).

**Matching Module.** From extracted features $(\hat{\mathbf{X}}_{\mathbf{a}}, \mathbf{F}_a)$ and $(\hat{\mathbf{X}}_{\mathbf{g}}, \mathbf{F}_g)$, we predict matched subsets $\mathbf{X}_{\mathbf{a}} \in \mathbb{R}^{n \times 2}$ and $\mathbf{X}_{\mathbf{g}} \in \mathbb{R}^{n \times 2}$. The matching is optimized to have $\mathbf{X}_{\mathbf{a}}$ and $\mathbf{X}_{\mathbf{g}}$ correspond to the same point. We utilize an attention-based graph neural net (GNN) that tackles partial matches and occlusions well using an optimal transport formulation, following Sarlin *et al.* [52]. The above neural feature extractor and GNN-based matcher help enjoy benefits of learning-based methods, particularly, those *pretrained* on large offline visual data without needing online, end-to-end finetuning. The geometric components, relying on these neural features, are described next.

**Lifting Points from 2D→3D.** Next, the agent's 2D local features are lifted to 3D with respect to the agent's coordinate frame *i.e.* $\mathbf{P}_a \in \mathbb{R}^{n \times 3}$. This is done by utilizing the camera intrinsic matrix $\mathbf{K}$

(particularly, principle point $p_x, p_y$ and focal lengths $f_x, f_y$) and the corresponding depth values for each position in $\mathbf{X_a}$, say $\mathbf{d_a} \in \mathbb{R}^n$. The $i^{\text{th}}$ row of $\mathbf{P}_a$ is calculated as

$$\mathbf{P}_a\,(i,:) = \left( \frac{\mathbf{X_a}(i,1) - p_x}{f_x} * \mathbf{d_a}(i), \frac{\mathbf{X_a}(i,2) - p_y}{f_y} * \mathbf{d_a}(i), \mathbf{d_a}(i) \right), \qquad (1)$$

where $\mathbf{X_a}(i,1)$ and $\mathbf{X_a}(i,2)$ correspond to the $x$ and $y$ coordinate of $i^{\text{th}}$ feature in $\mathbf{X_a}$, respectively. Formally, $\mathbf{d_a}(i) := \mathbf{D}_a\left(\mathbf{X_a}(i,1), \mathbf{X_a}(i,2)\right)$.

**Perspective-n-Point.** The objective of the next step *i.e.* Perspective-n-Point (PnP) is to find the rotation and translation between the agent and goal camera pose that minimized reprojection error. Concretely, for a given rotation matrix $\mathbf{R} \in \mathbb{R}^{3 \times 3}$ and translation vector $\mathbf{t} \in \mathbb{R}^3$, the 3D positions $\mathbf{P}_a$ of local features can be reprojected from the coordinate system of the agent to that of the goal camera:

$$\begin{bmatrix} \tilde{\mathbf{X}}_\mathbf{g} \\ 1 \end{bmatrix} = \mathbf{K}\,[\mathbf{R}|\mathbf{t}] \begin{bmatrix} \mathbf{P}_a \\ 1 \end{bmatrix}; \qquad \text{Reprojection error } e = \|\tilde{\mathbf{X}}_\mathbf{g} - \mathbf{X_g}\|_2^2. \qquad (2)$$

where $\tilde{\mathbf{X}}_\mathbf{g}$ are the reprojected positions. Minimizing the reprojection error $e$, via ePnP [51] and RANSAC [49] (to handle outliers), we obtain the predicted rotation and translation. The reprojection is visualized in Fig. 2, where the agent's amber point is lifted and reprojected in goal camera coordinates. The reprojection is different from its correspondence in the goal image.

**Estimating Distance and Heading to Goal.** The predicted translation $\mathbf{t}$ can help calculate the distance $\rho = \|\mathbf{t}\|_2$ from the agent to the goal. Similarly, the heading $\phi$ from the agent to the goal can be obtained from the dot product of the unit vectors along the optical axis (of the agent's view) and $\mathbf{t}$. Concretely, $\phi = \text{sgn}(t[1]) * \arccos\left(\mathbf{t} \cdot \mathbf{o}_a / \|\mathbf{t}\|_2 \|\mathbf{o}_a\|_2\right)$. The sign comes from $t[1]$ which points along the axis perpendicular to the agent's optical axis but parallel to the ground. The sign is particularly important when calculating the heading as it distinguishes between the agent turning right or left.

**Local policy.** Finally, the distance $\rho$ and heading $\phi$ between the agent's current position to the estimated goal are utilized to estimate actions in the action space $\mathcal{A}$ to reach the goal. Following accurate implementations [66, 1], we adopt a local metric map to allow the agent to heuristically avoid obstacles and move towards the goal. For further details, see Appendix A.

### 3.4 Switches

We define simple but effective switches between the two phases of goal discovery (*explore*) and last-mile navigation (*exploit*). The *explore→exploit* switch is triggered if the number of correspondences $n > n_{\text{th}}$, where $n_{\text{th}}$ is a set threshold. This indicates that the agent's image has significant overlap with the image-goal, so control can flow to the last-mile navigation phase. We find that this simple switch performs better than training a specific deep net to achieve the same (variations attempted in [1, 3, 4]). For *exploit→explore*, if the optimization for $\mathbf{R}, \mathbf{t}$ (see Eq. (2)) fails or if the predicted distance is greater than $d_{\text{th}}$ (tuned to 4m), the agent returns to the goal discovery phase.

## 4 Experiments

We report results for image-goal navigation both in photorealistic simulation and real-world scenes.

### 4.1 Data and Evaluation

We evaluate image-goal navigation policies on the benchmark introduced by Hahn *et al.* [1] and follow their evaluation protocol and folds. The benchmark consists of numerous folds: {Gibson [67], MP3D [68]}×{straight, curved}×{easy, medium, hard}. For a direct comparison to prior work [3, 1, 5, 6] that reports primarily on 'Gibson-curved' fold, we follow the same in the main paper. Consistent performance trends are seen in 'Gibson-straight' and in the MP3D folds as well. These results are deferred to Appendix C and Appendix F. Performance on image-goal navigation is chiefly evaluated via two metrics – percentage of successful episodes (*success*) and success weighted by inverse path length (*SPL*) [15]. For top-performing baselines, we also include the average distance to the goal at the end of the episode in Appendix G. The objective of the image-goal navigation task is to execute `stop` within $1m$ of the goal location. The agent is allowed 500 steps.

Table 1: **Results for 'Gibson-curved' episodes** Note the significant gains by adding SLING to prior works. Consistent trends are seen in 'Gibson-straight' (Appendix C) and MP3D-curved episodes (Appendix F).

|    | Method | Overall | | Easy | | Medium | | Hard | |
|----|--------|---------|---------|------|------|--------|------|------|------|
|    |        | Succ↑ | SPL↑ | Succ↑ | SPL↑ | Succ↑ | SPL↑ | Succ↑ | SPL↑ |
| 1  | BC w/ Spatial Memory [69] | 1.3 | 1.1 | 3.1 | 2.5 | 0.8 | 0.7 | 0.2 | 0.1 |
| 2  | BC w/ GRU [69, 70] | 1.7 | 1.3 | 3.6 | 2.8 | 1.1 | 0.9 | 0.5 | 0.3 |
| 3  | DDPPO [11] (from [1]) | 15.7 | 12.9 | 22.2 | 16.5 | 20.7 | 18.5 | 4.2 | 3.7 |
| 4  | NRNS [1] | 21.7 | 8.1 | 31.4 | 10.7 | 22.0 | 8.2 | 11.9 | 5.4 |
| 5  | ZER [5] | 33.0 | 23.6 | 48.0 | 34.2 | 36.0 | 25.9 | 15.1 | 10.8 |
| 6  | OVRL [6] | 45.6 | 28.0 | 53.6 | 31.7 | 47.6 | 30.2 | 35.6 | 21.9 |
| 7  | DDPPO-LMN + OVRL-GD | 44.3 | 30.1 | 52.4 | 36.6 | 48.6 | 32.6 | 31.9 | 21.2 |
| 8  | SLING + Straight-GD | 31.0 | 12.8 | 39.2 | 14.3 | 33.0 | 14.3 | 21.0 | 9.9 |
| 9  | SLING + DDPPO-GD | 37.9 | 22.8 | 52.2 | 32.7 | 42.2 | 25.2 | 19.4 | 10.5 |
| 10 | SLING + NRNS-GD | 43.5 | 15.1 | 58.7 | 17.4 | 47.0 | 17.4 | 25.0 | 10.5 |
| 11 | SLING + OVRL-GD | **54.8** | **37.3** | **65.4** | **45.7** | **59.5** | **40.6** | **39.6** | **25.5** |

## 4.2 Methods

We compare our last-mile navigation with several standardized baselines [69, 11, 1]. Note that field-of-view, rotation amplitude, *etc*. vary across baselines and we adopt the respective settings for fair comparison (implementation details of SLING are in Appendix A). Prior methods use a mix of sensors including RGB, depth, and agent pose, but no dense displacement vector to the goal. While we did include the most relevant baselines in Sec. 3.2, we also compare SLING to several other image-goal solvers. This includes imitation learning baselines such as Behavior Cloning (BC) w/ Spatial Memory [69] and BC w/ Gated Recurrent Unit [69, 70]. We also compare to established reinforcement learning baselines – DDPPO [11] and Offline Visual Representation Learning (OVRL) [6]. OVRL also makes use of pretraining using a self-supervised objective. Finally, we compare to related modular baselines include NRNS [1] and Zero Experience Replay (ZER) [5]. We defer a detailed discussion of these baselines to Appendix B.

**SLING & Ablations.** For a comprehensive empirical study, we combine SLING with Straight-GD, NRNS-GD, DDPPO-GD, OVRL-GD, and Oracle-GD (see Sec. 3.2 for details). We also introduce a neural baseline, DDPPO-LMN, a DDPPO model trained to perform last-mile navigation.

Further, we include clear ablations to show the efficacy of the components of our method and robustness to realistic pose and depth sensor noise:
• *w/ MLP switch:* instead of SLING's explore→exploit switch (that utilizes geometric structure), if a MLP[1] detects similarity between the agent and goal images (as in [1]).
• *w/o Recovery:* if the *exploit→explore* switch is removed *i.e.* one-sided flow of control.
• *w/o Neural Features:* if the neural features [65] are replaced with traditional features [71].
• *w/ Pose Noise:* add noise to pose that emulates real-world sensors [66, 72] (same as [3, 1]).
• *w/ Depth Noise:* imperfect depth by adopting the Redwood Noisy Depth model [73] in AI Habitat.
• *w/ Oracle-GD:* privileged baseline where NRNS-GD can access ground-truth distances to move the agent closer to the goal during exploration (see Sec. 3.2 and Appendix B).
• *w/ Oracle-LM-Pose:* privileged last-mile system with perfect displacement from agent to goal
• *w/ Oracle-LP:* privileged baseline where local policy can teleport agent to the goal prediction

## 4.3 Quantitative Results

In the following, we include takeaways based on the results in Tab. 1 and Tab. 2.

**State-of-the-art performance.** As Tab. 1 details, SLING + OVRL-GD outperforms a suite of IL, RL, and neural modular baselines. The Gibson-curved fold is widely adopted in prior works and hence the focus of the main paper. With a 54.8% overall success and 37.3 SPL we are the best-performing method on the benchmark, improving success rate by 21.8% *vs*. ZER and 9.2% *vs*. OVRL ('overall success' column of rows 5, 6, & 11). In Appendix I, we also demonstrate state-of-the-art performance when panoramic images are used.

---

[1]trained over an offline dataset of expert demonstrations, where adjacent nodes in a topological graph (that they maintain) are considered positives

Table 2: **Ablations on 'Gibson-curved' episodes.** Both switches are key to SLING's performance. SLING is resilient to sensor noise. Similar trends can be observed over ablations performed with OVRL-GD in Appendix F. The privileged last-mile navigation system establishes an upper bound for last-mile navigation. Even with Oracle-GD, performance improves if SLING is added.

|   | Method | Overall | | Easy | | Medium | | Hard | |
|---|--------|---------|---------|------|------|--------|--------|------|------|
|   |        | Succ↑ | SPL↑ | Succ↑ | SPL↑ | Succ↑ | SPL↑ | Succ↑ | SPL↑ |
| 1 | NRNS [1] | 21.7 | 8.1 | 31.4 | 10.7 | 22.0 | 8.2 | 11.9 | 5.4 |
| 2 | SLING + NRNS-GD | 43.5 | 15.1 | 58.7 | 17.4 | 47.0 | 17.4 | 25.0 | 10.5 |
| 3 | w/ MLP Switch | 42.5 | 14.8 | 55.4 | 16.7 | 47.3 | 17.3 | 24.9 | 10.5 |
| 4 | w/ MLP Switch w/o Recovery | 31.5 | 11.5 | 45.6 | 14.3 | 32.8 | 12.9 | 16.1 | 7.3 |
| 5 | w/o Neural Features | 33.7 | 11.3 | 47.5 | 13.5 | 35.9 | 13.0 | 17.7 | 7.5 |
| 6 | w/ Pose Noise | 43.7 | 14.3 | 58.6 | 16.1 | 47.6 | 16.8 | 24.9 | 10.1 |
| 7 | w/ Pose & Depth Noise | 43.5 | 14.0 | 56.9 | 15.9 | 47.2 | 15.9 | 26.6 | 10.3 |
| | *Privileged Last-Mile Navigation* | | | | | | | | |
| 8 | w/ Oracle-LP | 45.1 | 17.8 | 60.8 | 21.2 | 48.7 | 20.3 | 25.8 | 12.1 |
| 9 | w/ Oracle-LM-Pose | 53.3 | 19.3 | 72.3 | 23.4 | 57.1 | 21.6 | 30.5 | 13.1 |
| 10 | w/ Oracle-LM-Pose & Oracle-LP | 53.7 | 22.4 | 72.6 | 27.7 | 57.6 | 24.7 | 31.0 | 14.9 |
| | *Privileged Goal Discovery* | | | | | | | | |
| 11 | NRNS + Oracle-GD *(upper bound)* | 67.7 | 60.2 | 68.5 | 58.4 | 71.2 | 63.7 | 63.5 | 58.7 |
| 12 | SLING + Oracle-GD *(upper bound)* | 86.2 | 74.8 | 85.9 | 72.2 | 88.6 | 77.7 | 84.3 | 74.6 |

**SLING works across methods.** Using switches, we add our last-mile navigation system to DDPPO [11], NRNS [1], and OVRL [6], and observe gains across the board. As shown in Tab. 1, SLING improves the success rate of DDPPO by 22.2%, NRNS by 21.8%, and OVRL by 9.2% (rows 3 & 9, 4 & 10, 6 & 11, respectively). Quite surprisingly, SLING even with simple straight exploration, can outperform deep IL, RL, and modular baselines. (rows 1, 2, 3, 4, & 8).

**SLING outperforms neural policies for last-mile navigation.** SLING surpasses DDPPO trained over 400M steps for last-mile navigation by 10.5% on success rate (rows 7 & 11).

**SLING succeeds across scene datasets.** Similar improvements are also seen in MP3D scenes – adding SLING to OVRL improves success by 5.1%. Further details and results can be found in Appendix F.

**SLING is resilient to sensor noise.** As shown in rows 6 & 7 of Tab. 2, minor drops in performance are observed despite challenging noise in pose and depth sensors – SPL successively reduces 15.1→14.3→14.0% (rows 2→6→7).

**Geometric switches are better.** Performance reduces if we swap out SLING's explore→exploit switch with the MLP switch of NRNS [1]. The effect is exasperated when SLING's exploit→explore switch is also removed, leading to a drop of 12% (Tab. 2, rows 2 & 4). The neural features utilized in SLING are useful, as seen by comparing rows 2 and 5. Further, over a set of 6500 image pairs, we evaluate the accuracy of switches. SLING's explore→exploit switch is 92.0% accurate and MLP switch [1] is only at 82.1%. Also, SLING exploit→explore switch is 84.1% accurate while NRNS doesn't have such a recovery switch (details of this study in Appendix D).

**Large potential for last-mile navigation.** When Oracle-LM-Pose and Oracle-LP are used there is a 10.2% overall improvement in success from 43.5 to 53.7% (Tab. 2, rows 2 & 10). Notably, in easy episodes, oracle performance is an ambitious upper bound with an increase in success of 13.9% (58.7→72.6%). For the hard (*i.e.* longer) episodes, the oracle components have a relatively lower impact. This is quite intuitive as goal discovery errors are a more prominent error mode in long-horizon episodes instead of last-mile navigation.

**Improvements with Oracle-GD.** Even if we assume a perfect variant of goal discovery system from [1], we observe that performance saturates at 67.7% success (row 11, Tab. 2). Comparing rows 11 and 12, SLING can boost this asymptotic success rate by 18.5% (67.7→86.2%).

**Analysis: Why is SLING more robust?** In Fig. 3a, we visualize the frequency distribution of heading (from the agent to the target) in expert demonstrations [1] ('train GT') and that observed at inference ('test GT'). With no geometric structure, NRNS picks up the bias in training data, particularly, towards the heading of 0 (optimal trajectories entail mostly moving forward). Concretely, 72.2% of the training data is within $[-15°, 15°]$. This drops to 39.4% at test time when the last-mile

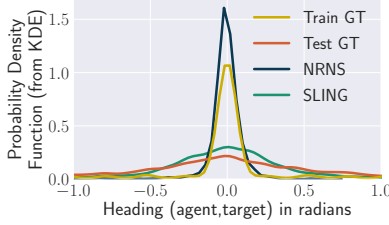

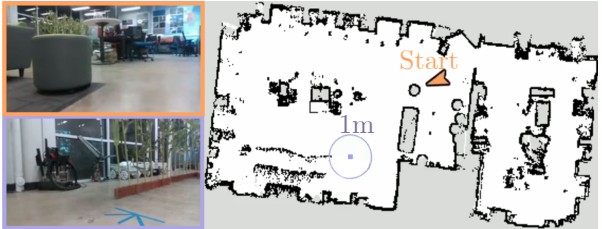

| (a) Bias in heading distribution | (b) Robot experiments (more visuals in Figures 5, 6, 7) |

Figure 3: (a) Significant distribution shift between training and test heading from agent to goal (Sec. 4.3), (b) Navigation policies deployed on a robot in cluttered real-world scenes (Sec. 4.4).

navigation phase is reached (using the best-performing Oracle-GD). Quantified with (first) Wasserstein distance, $W(\text{Test GT}, \text{NRNS}) = 0.0134$ *vs.* $W(\text{Test GT}, \text{SLING}) = 0.0034$, demonstrating SLING can better match the distribution at inference.

## 4.4 Physical Robot Experiments

We test the navigation policies on a TerraSentia [74] wheeled robot, equipped with an Intel® RealSense™ D435i depth camera (further hardware details in Appendix E). The robot is initialized in an unseen indoor environment and provided an RGB image-goal. We ran a total of 120 trajectories, requiring 30 human hours of effort, across three scenes and two levels of difficulty. Following the previously collected simulation dataset [1], easy goals are 1.5-3m from the starting location and hard goals are 5-10m from the starting location. Particularly, we test within an office and the common areas in two department buildings, over easy and hard episodes (following definitions from [1]). The physical setup (office) is shown in Fig. 3b. As in simulation, the agent is successful if it executes `stop` action within 1m of the goal. Examples of the image-goal utilized in physical robot experiments and precautions taken are included in Appendix E.

As shown in Tab. 3, for sim-to-real experiments, we base the goal discovery system on the NRNS model. We choose NRNS as the authors published an instantiation trained exclusively on real-world trajectories, particularly, RealEstate10K [14] (house tours videos from YouTube).

In preliminary experiments, we verified that this NRNS instantiation outperformed its simulation counterpart. For a direct comparison, in SLING + NRNS-GD, we utilize the same goal discovery system but add our switching and last-mile navigation system (SLING) around it. With SLING, we improve performance from 40.0% success to 56.6%. The gains become more prominent as the task horizon increases, leading to an improvement in success rate from 3.3% to 20.0%. The large gains in hard episodes (which are exploration heavy) are accounted to SLING's better explore→exploit switch and SLING's last-mile navigation system that is not biased to zero heading (particularly important for curved and long episodes).

|        | Easy |  | Hard |  |
|--------|------|------|------|------|
| Method | Succ↑ | SPL↑ | Succ↑ | SPL↑ |
| NRNS [1] | 40.0 | 37.7 | 3.3 | 3.3 |
| + SLING | **56.6** | **53.7** | **20.0** | **19.3** |

Table 3: Results in real-world scenes.

## 5 Conclusion

In this work, we identify and leverage the geometric structure of last-mile navigation for the challenging image-goal navigation task [1]. With analysis of data distributions, we demonstrate that learning from expert demonstrations may lead to developing a bias. Being entirely complementary to prior work, we demonstrate that adding SLING leads to improvements across data splits, episode complexity, and goal discovery policies, establishing the new state-of-the-art for image-goal navigation [1]. We also transfer policies trained in simulation to real-world scenes and demonstrate significant gains in performance. Further improvements in the switching mechanism, neural keypoint features, visual representations from view augmentations, *etc.* complement our proposed approach to help improve performance in future work.

Like any method, SLING has several aspects where follow-up works can improve on. We list them explicitly: (1) Our method is limited by mistakes in matching correspondences. (2) We add additional parameters that need to be tuned. (3) We make a single prediction for last-mile navigation. (4) We assume access to depth and pose information. More details of these aspect as well as a discussion on pose errors, depth noise, and the nuanced image-goal navigation definition in Appendix J.

## Acknowledgments

We thank the reviewers for suggesting additional experiments to make the work stronger. JW and GC are supported by ONR MURI N00014-19-1-2373. We are grateful to Akihiro Higuti and Mateus Valverde for physical robot help, Dhruv Batra for helping broaden the scope, Jae Yong Lee for help with geometric vision formulation, Meera Hahn for assistance in reproducing NRNS results, and Shubham Tulsiani for helping ground the work better to 3D vision. A big thanks to our friends who gave feedback to improve the submission draft – Homanga Bharadhwaj, Raunaq Bhirangi, Xiaoming Zhao, and Zhenggang Tang,

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
