# OpenReview forum: "Last-Mile Embodied Visual Navigation"
_robot-learning.org/CoRL/2022/Conference — CoRL 2022 Poster_

### Official Review · Reviewer_wKSA · 2022-07-29

**Originality:** Good
**Technical Quality:** Good
**Clarity Of Presentation:** Very Good
**Impact:** 3

**Recommendation:**

Weak Reject: I recommend rejecting the paper, but will not argue for my recommendation if the majority of other reviewers have a different opinion.

**Summary:**

This paper addresses the challenge of last-mile goal-reaching in visual navigation tasks, i.e. the last stage of navigating to an image-goal in an unknown environment, when the goal is in the line-of-sight of the agent. The authors propose a simple controller that extracts visual keypoints in the agent’s observations and establishes correspondence to the goal image — converting the last-mile image-goal problem into a point-goal problem. Extensive quantitative analysis suggests that this indeed improves performance on benchmark tasks, and the learned policies generalize to real-world indoor environments.


**Issues:**

- [Points Raised Above] I raised some concerns in the above section and some clarification in response to those points would really help me re-evaluate the submission score.
	- Adding new baselines for upper bound and a better metric
	- Understanding Oracle-GD baseline


**Quality Of The Limitations Section:**

Limitations are addressed clearly

**Reviewer Expertise:**

5: The reviewer is absolutely certain that the evaluation is correct and very familiar with the relevant literature

**Robotics Focus:**

Sufficient demonstration on hardware

**Strengths And Weaknesses:**

Strengths
---
- The paper is very well-written and the observation of converting the last mile to image-goal navigation tasks to a point-goal navigation task, albeit simple, seems well motivated. I also found the method explanation to be easy to follow.
- The authors do a good job at statistically significant evaluations (both in sim and real) of the tasks and baselines presented. The experiments compare against SOTA visual navigation techniques and include useful ablations that make it very easy to understand which design decisions were most relevant.

***
Weaknesses
---
- [Brittle Components] While the proposed method makes complete sense on paper, I am very skeptical of the {feature extraction + correspondence matching + lifting + SFM + geometric estimation} pipeline being broadly usable beyond simulated Habitat tasks and simple indoor settings. The biggest motivation for learning-based approaches to navigation is to reduce the “hard” reliance of the geometric pipelines on factors such as camera intrinsic parameters, local odometry and pose estimation, etc. While re-combining learning and geometry is generally a good idea, the specific mechanism proposed in the paper relies a bit too heavily on the brittle parts of geometric method and it seems unlikely to broadly useful.
- [Interpreting Results: Upper Bound] I had some trouble interpreting the results at different environment complexities and what the contribution of the proposed method (SLING) really is — having some sort of an oracular upper bound baseline would be really useful here. The improvements in many cases seem absurdly high, and it is not clear why such large improvements are caused by SLING (esp. for real-world experiments); the paper does not discuss this too well. Two baselines I strongly recommend adding, that shed light on different aspects of the system, can be:
	-  An oracle baseline for the exploit-only phase would be a useful addition to have (the paper currently contains Oracle-GD, not in exploit case). This could serve as an _upper bound_ to evaluate how good the detection and PointGoal performance really is.
	- An evaluation with increased tolerance (for any baseline: NRNS/OVRL) that directly gets at the hypothesis presented to motivate the two-stage process (Line 73).
	- Another important metric to include (in Table 1) would be the “distance of trajectory endpoint to goal” for the non-SLING baselines. This would really help understand the failure modes better — are the failures happening mostly due to long-horizon issues in the last mile, sooner, or other reasons.
- [Understanding Oracle-GD Improvements] I don’t follow the argument about result improvements with Oracle-GD (Line 245). The *NRNS+Oracle-GD* baseline has access to the ground-truth environment state, then why does adding SLING to this baseline lead to such a massive jump in performance? Particularly, the goal-discovery phases of the two methods are identical, and post discovery, SLING only adds information about the estimated goal location but this is already available (noise-free) to the method — why then does the performance change at all? I may be misinterpreting how this baseline works, but this needs to be clarified in the text better.


**Summary Of Recommendation:**

In my review, I raised a major concern regarding the significance of the findings (owing to brittle components), and suggestions regarding presentation of experimental results. I cannot recommend acceptance of the paper in its current form, but I am very willing to engage with the authors in the discussion phase and reconsider my recommendation.

---
**Update:** After the round of discussions with the authors, my concerns have been mostly addressed, as long as the authors actually incorporate these (and other changes/additional expts requested by other reviewers) changes in the final version. I'm on the borderline between a WA/WR and cannot enthusiastically recommend acceptance, but I have no major actionable concerns with the paper after the recent discussion.

---

> ### Author Response · Authors · 2022-08-24
> **Response to Reviewer wKSA (Part 1)**
>
> We thank the reviewer for recognizing the easy-to-follow writing, statistically significant evaluations across simulation and real, effective baselines, and clear ablations in our work. We are happy to see the reviewer is open to engaging and reconsidering their recommendation. We’ve included results for all requested experiments. We hope these results and insights further cement the robustness of SLING. We include answers and results next.
> > While the proposed method makes complete sense on paper, I am very skeptical of the {feature extraction + correspondence matching + lifting + SFM + geometric estimation} pipeline being broadly usable beyond simulated Habitat tasks and simple indoor settings.
>
> __SfM__: There is __some misinterpretation__ here. We do not employ or mention structure from motion (SfM) in our paper. We cast the goal camera estimation problem as a perspective-n-point formulation (estimation of camera pose from 3D points in the world and 2D projections in an image). In real-robot experiments, we use a very robust SLAM method (RTAB-Map [7], ORB-SLAM[8]) for pose estimations that fill in for the freely-available pose in the ImageNav task definition within AIHabitat.
>
> __Simple indoor settings__: To the best of our knowledge, we are the first to demonstrate ImageNav policies transfer from photorealistic simulation to physical settings, needing _no online training in the physical world_. The three environments we test in are very challenging -- they include diverse layouts and furniture, several obstacles, different light conditions, long hallways, and visually-confounding common spaces (due to repeated patterns).
>
> __Broadly useful__: We have concretely provided evidence for SLING’s broader applicability in the context of the current state-of-the-art in learning-based robotics by comparing to four published baselines spanning IL, RL, and neural modular policies (Tab. 1). Moreover, SLING is simple and easy to deploy with just 100 lines of additional code (in supplement). Finally, we show SLING improves performance across six episode complexities, two scene datasets, and a zero-shot sim-to-real transfer to challenging physical settings (Tab. 3). All components are rigorously evaluated in an ablation (Tab. 2). We strongly believe that simple and principled SLING improving all traditional monolithic policies for ImageNav is a useful inference for our research community.
>
> __Very skeptical of the pipeline__: We first point out that neural feature-based correspondences with SLAM are one the most stable pipelines deployed in the real world. Further, given the extensive empirical evidence that we provide, _we see no objective reason for skepticism towards our method_. We earnestly provide all additional experiments that were requested in the review. We are very happy to answer any other follow-ups that the reviewer finds will help them reconsider their recommendation.
> ***
> > The biggest motivation for learning-based approaches to navigation is to reduce the “hard” reliance of the geometric pipelines on factors such as camera intrinsic parameters, local odometry and pose estimation, etc. While re-combining learning and geometry is generally a good idea, the specific mechanism proposed in the paper relies a bit too heavily on the brittle parts of geometric method and it seems unlikely to broadly useful.
>
> A few clarifications about the functionality of SLING that might be helpful here:
>
> 1. __SLING is also learning-based__: We duly employ learning where it has shown promise in the literature. Note that we extract neural local features and utilize deep graph neural nets for correspondence matching. The key difference is that these networks are _pretrained_ on offline data as opposed to _finetuned_ end-to-end.
>
> 2. __We add just three easy-to-design parameters__: We include these in L440-444 in the supplement. Our method isn’t brittle. In stark contrast, our intuitive hyperparameter choices work across all four baselines (Tab. 1) and are resilient to noise in both pose and depth sensor noises (rows 6 and 7 of Tab. 2). No automated or grid-search tuning was conducted.
>
> 3. __Local odometry and pose estimation__: Several methods demonstrating sim-to-real transfer for navigation in the real world have found pose estimations and SLAM to be very effective [9, 10, 11]. From our experience too, we found SLAM to be robust across scenes.
>
> 4. __Camera intrinsics__: Utilizing camera intrinsics is a mathematically rigorous operation grounded in optics. Empirically too, we found absolutely no issues deploying SLING across two different cameras (AIHabitat setup of 90°/120° FoV camera and the robot’s Intel® RealSense™ D435i with a horizontal FoV of 69° and vertical FoV of 42°).

---

> > ### Author Response · Authors · 2022-08-24
> > **Response to Reviewer wKSA (Part 2)**
> >
> > > An oracle baseline for the exploit-only phase would be a useful addition to have
> >
> > This is a great suggestion. To realize it, we sequentially replaced each component of SLING with its oracle counterpart. Particularly, (1) _oracle goal detection_: access to ground truth translation and heading from agent to goal (2) _oracle pointgoal navigation_: this teleports the agent to a given goal prediction. The results are included below
> >
> > |{$\cdot$} + NRNS-GD [1] |Overall Success ↑|Overall SPL ↑|Easy Success ↑|Easy SPL ↑|Medium Success ↑|Medium SPL ↑|Hard Success ↑|Hard SPL ↑|
> > |--------------------|------------|-----------|---------|--------|--------|-------|---------|--------|
> > |SLING               |43.5        |15.1       |58.7     |17.4    |47.0      |17.4   |25.0       |10.5    |
> > |SLING + oracle pointnav |45.1        |17.8       |60.8     |21.2    |48.7    |20.3   |25.8     |12.1    |
> > |SLING + oracle detection |53.3        |19.3       |72.3     |23.4    |57.1    |21.6   |30.5     |13.1    |
> > |SLING + oracle pointnav + oracle detection     |53.7        |22.4       |72.6     |27.7    |57.6    |24.7   |31.0       |14.9    |
> >
> > _Analysis_: These results show that on easy episodes, oracle performance is an ambitious upper bound. Improvements in goal detection and navigation to this goal prediction should help bridge this gap to the upper bound.
> > For the hard (i.e. longer) episodes, the oracle components have a relatively lower impact. This is quite intuitive as _exploration_ (or goal discovery) errors are a more prominent error mode in long-horizon episodes instead of last-mile navigation.
> > ***
> > > An evaluation with increased tolerance (for any baseline: NRNS/OVRL) for L73.
> >
> > This is a valuable experiment to substantiate our motivation, akin to [4, 5, 6]. In this new experiment, following the corresponding study for multi-object navigation [5], we increase the tolerance of the ‘stop’ action for the NRNS baseline. This _stop budget_ allows the agent to continue last-mile navigation beyond a ‘hard’ failure till the budget is exhausted. With just a budget of one, success increased from 28 to 51%. The smooth increase in performance with increase in tolerance towards last-mile navigation errors (as realized by culminating actions) is consistent with prior work (particularly the Tab. 9 of the supplement in MultiON [5]). This shows that improving the last-mile of navigation and recovering from mistakes has great potential. This is the precise focus of our switches and last-mile navigation module.
> >
> > |STOP budget         |Overall Success ↑|Overall SPL ↑|
> > |--------------------|------|-----|
> > | 0 (from [1])       |21.8  |8.1  |
> > | 0 (reproduced*)    |27.8  |10.7 |
> > |1                   |50.8  |17.0   |
> > |2                   |68.4  |21.4 |
> > |3                   |81.2  |25.1 |
> > |4                   |90.9  |28.3 |
> > |5                   |96.2  |30.0   |
> > |6                   |98.8  |31.0   |
> > |7                   |99.8  |31.2 |
> > |8                   |100   |31.3 |
> >
> > \* We tweak NRNS implementation slightly to prevent redundant nodes from being added to the topological map. This leads to clear gains at no cost.
> > ***
> > > Another important metric to include … “distance of trajectory endpoint to goal”
> >
> > We embrace this suggestion to provide another axis of comparison for the baselines and SLING. We report this metric below for the top-performing methods:
> >
> > | Method | Overall Final Dist. ↓ | Easy Final Dist.  ↓ | Med Final Dist.  ↓ | Hard Final Dist. ↓  |
> > |-------------------|------------------|---------------|--------------|---------------|
> > | DDPPO [3]            | 3.05             | 2.28          | 2.75         | 4.11          |
> > | DDPPO-GD + SLING       | 2.61             | 1.68          | 2.18         | 3.96          |
> > | NRNS [1]             | 2.96             | 1.99          | 2.74         | 4.15          |
> > | NRNS-GD + SLING        | 2.42             | 1.41          | 2.10          | 3.75          |
> > | OVRL [2]             | 2.43             | 1.58          | 2.12         | 3.59          |
> > | OVRL-GD + SLING        | __2.17__             | __1.28__          | __1.70__          | __3.52__          |
> >
> > Analysis:
> >
> > * Across DDPPO, NRNS, and OVRL the following trends hold: (1) the final distance to goal is much lower than the initial distance to the goal, (2) the final distance to goal is within range of last-mile navigation, (3) SLING significantly reduces final distance to goal.
> >
> > * Elaborating on (1), stated in averages, base OVRL starts ~2.25m from the goal in easy episodes (1.5-3m) and reaches 1.58m from it, starts ~4m from the goal in medium episodes (3-5m) and reaches 2.12m from it, and starts ~7.5m from the goal in hard episodes (5-10m) and reached 3.59m from it. As we detail in L493-498, we intuitively find distances of <3m can count towards last-mile navigation.

---

> > > ### Author Response · Authors · 2022-08-24
> > > **Response to Reviewer wKSA (Part 3)**
> > >
> > > > Understanding Oracle-GD Improvements … I may be misinterpreting how this baseline works, but this needs to be clarified in the text better.
> > >
> > > We thank the reviewer for this feedback and we will revise. We believe the following should clarify the nuanced oracle-GD ablation:
> > >
> > > 1. _Oracle-GD is an oracle explorer, but not an oracle policy_: As we include in L122 and L468, oracle-GD is NRNS-GD [1] but with direct access to ground-truth geodesic distances during the goal discovery phase. Delving into some details, (NRNS/oracle)-GD chooses a node (from an accumulated topological map) that it estimates to be closest to the goal and navigates to it. Ground-truth distances in oracle-GD imply that the module will choose the optimal node, i.e., the agent will deterministically move closer to the goal during exploration. It does not have any other privileged information i.e. it isn’t an oracle ImageNav policy, as detailed next.
> > >
> > > 2. _The switch isn’t oracle_: While oracle-GD brings the agent closer to the goal, the switch (from goal discovery to last-mile navigation) is not perfect. Either the MLP switch of NRNS (L210) or SLING’s switch are employed in _{$\cdot$} + Oracle-GD_ baselines; both are imperfect.
> > >
> > > 3. _Last-mile navigation is still error-prone_: Now, after the switch, the responsibility of calling ‘stop’ and navigating the last mile still belongs to either SLING or the ‘target prediction network’ of NRNS in _{$\cdot$} + Oracle-GD_ baselines.
> > >
> > > 4. _“SLING only adds information about the estimated goal location but this is already available (noise-free) to the method”_: We gently state that this is incorrect. The only privileged information available to oracle-GD [1] is ‘distance’ (not even the displacement vector) from a given node to the goal, to choose a node close to the goal for exploration. Critically, this privileged information is used exclusively in the exploration module, *not* in the exploitation module. SLING’s information about translation and orientation to the goal is still necessary for effective navigation of the last mile.
> > >
> > > In this oracle goal discovery setting, NRNS’s attempt at last-mile navigation leads to 67.7% success while SLING leads to a significantly higher success rate of 86.2% (rows 8 and 9 of Tab. 2).
> > >
> > > __References__:
> > > [1] M. Hahn et al. No rl, no simulation: Learning to navigate without navigating. In NeurIPS, 2021.
> > > [2] K. Yadav et al. Offline visual representation learning for embodied navigation. arXiv preprint arXiv:2204.13226, 2022.
> > > [3] E. Wijmans et al. Dd-ppo: Learning near-perfect pointgoal navigators from 2.5 billion frames. In ICLR, 2019.
> > > [4] P. Chattopadhyay, J et al. Robustnav: Towards bench-marking robustness in embodied navigation. In ICCV, 2021.
> > > [5] S. Wani et al. Multion: Benchmarking semantic map memory using multi-object navigation. In NeurIPS, 2020.
> > > [6] J. Ye et al. Auxiliary tasks and exploration enable objectgoal navigation. In ICCV, 2021.
> > > [7] M. Labbé et al. RTAB-Map as an Open-Source Lidar and Visual SLAM Library for Large-Scale and Long-Term Online Operation. In JFR, 2019.
> > > [8] R. Mur-Artal et al. ORB-SLAM2: an open-source SLAM system for monocular stereo and RGB-D cameras. In T-RO, 2017.
> > > [9] P. Anderson et al. Sim-to-real transfer for vision-and-language navigation. In CoRL, 2021.
> > > [10] J. Truong et al. Bi-directional domain adaptation for sim2real transfer of embodied navigation agents. In RA-L, 2021.
> > > [11] A. Kadian et al. Sim2real predictivity: Does evaluation in simulation predict real-world performance? In RA-L, 2020.

---

> ### Comment · Reviewer_wKSA · 2022-08-25
> **Reviewer Response**
>
> Thanks for the updated discussion and new experiments that significantly improve the analysis of contributions. This addresses most of my concerns around evaluation and presentation.
>
> Regarding the related work discussion, I should emphasize that the "ImageNav" task has been well-studied in prior work, especially with real robots. The current section has a discussion of simulation/Habitat-centric approaches to ImageNav, but this paints a very incomplete picture of progress in the field. Also, the authors mention "we are the first to demonstrate ImageNav policies transfer from photorealistic simulation to physical settings, needing no online training in the physical world", which is not true. See [Meng et al.](https://arxiv.org/abs/1909.12329). The section should also be expanded to acknowledge prior work like [Hirose et al. 2018](https://arxiv.org/abs/1903.02749), [Shah et al. 2020](https://arxiv.org/abs/2012.09812), [Meng et al. 2019](https://arxiv.org/abs/1909.12329), [Shah et al. 2022](https://arxiv.org/abs/2202.11271).
>
> I would also like to see the promised changes and updated discussion reflected in a revised PDF.

---

> > ### Author Response · Authors · 2022-08-27
> > **Thanks and revised PDF**
> >
> > Indeed, the reviewer’s suggestions have improved the evaluation and presentation of our work. Thank you!
> >
> > Regarding the related work discussion: We’ve integrated all suggested references. In particular, we’ve broadened the image-goal navigation subsection to connect our work to indoor and outdoor robot navigation.
> >
> > Regarding the request for a revised PDF: Given the tight timeline of the author response period (ends 27th Aug), we have made changes on an expedited timeline (attached). Revisions specific to the reviewer are highlighted in _purple_. Once OpenReview begins accepting revisions, at a prospective camera-ready stage, we’ll update the main PDF.
> >
> > Given the fruitful discussion, we sincerely hope the reviewer will reconsider their earlier recommendation. If the reviewer has any remaining questions or follow-ups, we will be prompt to respond. Thanks again for engaging with us during this period.

---

### Official Review · Reviewer_FqFC · 2022-07-31

**Originality:** Fair
**Technical Quality:** Excellent
**Clarity Of Presentation:** Very Good
**Impact:** 3

**Recommendation:**

Weak Accept: I recommend accepting the paper, but will not argue for my recommendation if the majority of other reviewers have a different opinion.

**Summary:**

This paper tackles the problem of last-mile embodied visual navigation, in which the agent has to navigate to the goal after the goal image is visible in the current view. Once the image is visible, the correspondences between the current and the goal image is used to recover the relative rotation and translation of the two images. This, combined with the depth, is used to estimate the heading and distance to the goal, which is then used with a local metric map to avoid obstacles to head to the given goal. The paper also proposes ways to switch between goal discovery and last mile navigation phases, enabling the proposed method to easily be combined with existing goal discovery methods. The method is evaluated thoroughly on Gibson and MP3D datasets with numerous state of the art baselines establishing that the proposed method performs much better than the baselines and gets consistent improvement.

**Issues:**

From the comments above:

1. Lacks an alternative baseline method comparison for last mile navigation itself. One baseline which would be good to have is to learn a policy to navigate to the goal when initialized with the last mile navigation task, i.e., goal in sight.
2. Another baseline that would be instructive to have is Neural Topological SLAM [3], which also seems to use the geometric information which should implicitly lead to good last mile navigation performance.
3. In L273-274, the authors say that the gains become sharper for the hard tasks. For completeness, could the authors describe what is the difference in these tasks compared to the easier benchmark in the real world? If the task just entails more distant goals, then I would assume that this is the job of goal discovery, and not last mile navigation, and hence would require an explanation on why the performance of the proposed method is significantly better in this regime.


**Quality Of The Limitations Section:**

Limitations are addressed clearly

**Reviewer Expertise:**

3: The reviewer is fairly confident that the evaluation is correct

**Robotics Focus:**

Sufficient demonstration on hardware

**Strengths And Weaknesses:**

1. The paper is well motivated. It picks the task of last mile navigation which empirically causes a non-trivial loss in performance in the proposed end-to-end navigation methods.
2. The paper is well written, has illustrative figures, and is easy to follow.
3. The experiments and ablations are quite thoroughly done on both Gibson, MP3D and real-world scenes. The baselines look relevant and include state-of-the-art methods including end-to-end RL, Imitation Learning, and Neural Modular Policies. The improvements convincingly show that the proposed method SLING adds non-trivial gains when combined with the existing goal discovery methods.
4. A limitation of the proposed method is that it has several error-prone steps and hyperparameters which make it very complicated to implement and deploy. It would be good to simplify some of the steps by subsuming them in a learned policy, as discussed below in the suggested baseline.
5. Another limitation of the paper is that the paper compares with no alternative baseline methods for last mile navigation itself. One baseline which would be good to have is to learn a policy to navigate to the goal when initialized with the last mile navigation task, i.e., goal in sight.
6. Another baseline that would be instructive to have is Neural Topological SLAM [3], which also seems to use the geometric information which should implicitly lead to good last mile navigation performance.
7. In L273-274, the authors say that the gains become sharper for the hard tasks. For completeness, could the authors describe what is the difference in these tasks compared to the easier benchmark in the real world? If the task just entails more distant goals, then I would assume that this is the job of goal discovery, and not last mile navigation, and hence would require an explanation on why the performance of the proposed method is significantly better in this regime.


**Summary Of Recommendation:**

The paper focuses on the problem of last-mile navigation and gives a compelling empirical case to show that the proposed method SLING can lead to consistent improvement across the board when combined with existing goal discovery methods. I'm leaning towards an acceptance because the paper has strong empirical evidence to support the importance of separately solving the last mile navigation task, and then combining it with existing goal discovery methods.

---

> ### Author Response · Authors · 2022-08-24
> **Response to Reviewer FqFC (Part 1)**
>
> Thank you for your positive feedback about our research objective, writing, thorough experimentation, and relevant baselines across RL, IL, and neural modular policies. We respond to your questions below:
> > Several error-prone steps and hyperparameters … complication to implement and deploy
>
> We request the reviewer to reconsider this inference based on the following four objective points:
>
>
> 1. Our implementation of SLING is _just ~100 lines of additional code_ (as included in our supplement: https://anonymous.4open.science/r/SLING) beyond the base policies it is added to. Existing baselines are completely reused for all but the switch and last-mile navigation. We believe this is a strong indicator of ease of implementation.
>
> 2. As for steps, we deliberately devised interpretable steps to build a principled last-mile navigation approach based on geometric 3D vision. In Tab. 1, we deployed SLING off-the-shelf with four diverse baselines, with no hyperparameter tuning or addition/removal of any of the steps and it significantly boosted metrics. This supports SLING’s ease of deployment.
>
> 3. As for hyperparameters, we have _exactly 3 new design parameters_ (detailed in L440-444 in the supplement). Note that we use different # of matches (50 in SLING + NRNS-GD vs. 20 in SLING + OVRL-GD) because the base methods NRNS/OVRL use different input image sizes.
> Our intuitive hyperparameter choices are resilient to noise in both pose and depth sensors noises (rows 6 and 7 of Tab. 2). No automated or grid-search tuning was conducted.
>
> 4. Finally, as reported in Sec. 4.4, our extensive physical experiments corroborate that SLING can be _taken to the real world without any online fine-tuning_. This is the first demonstration of off-the-shelf transfer for image-goal navigation in a challenging physical environment.
>
> We sincerely hope this convinces the readers, including the reviewer, of our motivation behind SLING – to build a simple and effective method that can be easily plug-and-played without tuning. As explained in L44, L293, and headings of Sec. 4.3, we directly demonstrate this by deploying SLING across four baselines, six episode complexities, two simulated scene datasets, and a zero-shot sim-to-real transfer to three challenging physical environments.
> ***
> > One baseline which would be good to have is to learn a policy to navigate to the goal when initialized with the last mile navigation task
>
> This is a great addition to our experiments. To realize this suggestion, we re-trained the end-to-end RL baseline of DDPPO [4] by initializing it exclusively for the last-mile navigation task (<3m from the goal). We denote this last-mile specific model as DDPPO-LMN and train it to convergence in 400M steps in the Gibson scenes. For a head-on comparison with SLING, we allow DDPPO-LMN to use the same switch as SLING. The results are included below.
>
> |Method          |Overall Success ↑|Overall SPL ↑|
> |----------------|-----------------|-------------|
> |OVRL            |45.6             |28.0           |
> |OVRL + DDPPO-LMN $\leftarrow$ _new_ |44.3             |30.1         |
> |OVRL + SLING    |__54.8__         |__37.3__     |
>
> Analysis: When used in conjunction with OVRL, DDPPO-LMN improves the overall SPL of OVRL i.e. more targeted path planning. SLING’s last-mile navigation does significantly better than base OVRL as well as the new baseline with the neural policy specific to last-mile navigation.

---

> > ### Author Response · Authors · 2022-08-24
> > **Response to Reviewer FqFC (Part 2)**
> >
> > > Another baseline that would be instructive to have is Neural Topological SLAM … seems to use the geometric information
> >
> > We agree that NTS is a relevant baseline. Let us first add some context before including relevant results:
> >
> > Quite early in the timeline of this project, we reached out to the authors of NTS [5]. They were unable to make their code public and encouraged us to benchmark to their open-sourced follow-up, NRNS [1]. NRNS has very similar components and is from largely the same research team. Importantly, NTS operates on an entirely different dataset that has not been released. Finally, NRNS does not provide an implementation or report metrics for NTS in their paper or code.
> >
> > In light of the above constraints, we still include _three new insights_ that can help the readers (including the reviewer) better compare NTS to SLING:
> > 1. The ‘relative pose prediction’ function is SLING’s counterpart in NTS. This function transforms two images to output their relative pose difference, for the agent to travel towards. Importantly, unlike SLING, this function _does not utilize geometric information_. This function is essentially the same as the ‘target prediction network’ in NRNS. Note, that we were able to improve NRNS success rate from 21.7% to 43.5% when using SLING instead of using the ‘target prediction network’ (SPL almost doubles with SLING: from 8.1% to 15.1%).
> >
> > 2. The ‘_geometric_ explorable area prediction’ function (dubbed $\mathcal{F}_{g}$) in NTS might be what you are referring to. The paper provides little in the way of the geometric information that it uses, just stating that the module is used to predict free space. Regardless, this _‘geometric’ function is purely used for exploration_. Progress in exploration is orthogonal to our novel research focus on last-mile navigation. We have provided empirical evidence of plug-and-playing SLING with four exploration baselines including neural modular policies like NRNS (NTS belongs to the same category).
> >
> > 3. NTS operates on panoramic observations. This is in sharp contrast to most other methods benchmarked in most prior works [2, 3, 4] and in ours that operate on FoV of 90° or 120° (Sec. 4.1 and L425-430 in supplement). To provide additional baselines in this new panoramic domain (of NTS), we retrained the top baselines from the main paper. Results are included below:
> >
> > |Method                  |Overall Success ↑|Overall SPL ↑|
> > |------------------------|---------------|-----------|
> > |OVRL (90° FoV)          |45.6           |28         |
> > |OVRL + SLING (90° FoV)  |__54.8__           |__37.3__       |
> > |OVRL (panoramic) $\leftarrow$ _new_        |76.7          |59.4      |
> > |OVRL + SLING (panoramic) $\leftarrow$ _new_ |__78.6__          |__60.4__      |
> >
> > We hope the unavailability of NTS implementation and dataset, the above insights, and new results help mitigate the concern of the reviewer about NTS.
> > ***
> > > L273-274 … difference between easy and hard benchmark in the real world
> >
> > This is very helpful feedback. In Sec. 4.3, we missed including that _easy_ and _hard_ trajectories are defined consistent with the corresponding simulation dataset [1]. We will revise to include that easy episodes are 1.5-3m from start to goal while hard episodes are 5-10m from start to goal.
> > ***
> > > Why is there more improvement in the hard tasks; isn’t that the job of goal discovery?
> > SLING tackles hard tasks better due to two reasons, that might not be apparent right away:
> > 1. __Switches__: Moving too early to exploitative last-mile isn’t too bad for easy episodes, but is catastrophic for hard ones. In physical experiments, we found SLING’s switches were accurate, typically only going into last-mile navigation in the vicinity of the goal. In contrast, one-sided MLP switch from NRNS has a high false-positive rate. This causes the robot to go from exploration to exploitation too early. This is consistent with our switch accuracy study in simulation (reported in L242-243): SLING switches were 92% accurate in the explore $\rightarrow$ exploit detections while the MLP switch was only 82.1% accurate.
> > 2. __Resilience to data bias__: As concretized in Fig. 3(a), NRNS has a faulty bias towards predicting a heading of 0 (as it learns from expert training demonstrations). On the other hand, SLING has no such bias. This enables the large improvements obtained with SLING. Note that the negative impact of zero-heading bias is more pronounced in ‘hard’ (i.e. 5-10m long) episodes than easy (i.e. 1.5-3m short episodes). Longer episodes necessitate more exploration $\Rightarrow$ broader test-time distribution of heading $\Rightarrow$ larger gap from the train distribution $\Rightarrow$ more room for SLING to improve.

---

> > > ### Author Response · Authors · 2022-08-24
> > > **References**
> > >
> > > [1] M. Hahn et al. No rl, no simulation: Learning to navigate without navigating. In NeurIPS, 2021.
> > > [2] K. Yadav et al. Offline visual representation learning for embodied navigation. arXiv preprint arXiv:2204.13226, 2022.
> > > [3] Z. Al-Halah et al. Zero experience required: Plug & play modular transfer learning for semantic visual navigation. In CVPR, 2022
> > > [4] E. Wijmans et al. Dd-ppo: Learning near-perfect pointgoal navigators from 2.5 billion frames. In ICLR, 2019.
> > > [5] D. S. Chaplot et al. Neural topological slam for visual navigation. In CVPR, 2020.

---

> ### Author Response · Authors · 2022-08-27
> **Rebuttal Summary and Updated Manuscript**
>
> From the review, we believe that the main questions/suggestions were:
>
> (1) Probable implementation complexity:
>
> Addressed by highlighting the ease of implementation (100 lines of additional code), ease of deployment (across diverse baselines), just three additional hyperparameters that are intuitively set, and sim-to-real transfer without any online finetuning in the real world.
>
> (2) Last-mile navigation-specific baseline:
>
> New experiments based on the suggested baseline reaffirm SLING’s efficacy. We’ve integrated this in Sec. 4.3 and the _new Appendix J_.
>
> (3) NTS comparison
>
> Addressed by adding the context of the unavailability of NTS implementation and dataset. In new experiments based on NTS’s panoramic observations, SLING improves over the prior state-of-the-art. We’ve integrated this in Sec. 4.3 and the _new Appendix I_.
>
> (4) Clarification about physical experiments and SLING’s improvement in hard episodes
>
> Addressed by clarifying easy and hard episode definitions. _Better switches_ and _resilience to data bias_ help SLING improve hard episodes significantly. We’ve integrated this in Sec. 4.4.
>
> We believe that we concretely address all concerns shared in the review. Is our response and updated manuscript (reviewer-specific edits are in _green_) satisfactory? If not, please let us know how we can better address these concerns.
>
> Thank you so much for your efforts to help improve our work!

---

### Official Review · Reviewer_D7Jh · 2022-08-02

**Originality:** Good
**Technical Quality:** Good
**Clarity Of Presentation:** Good
**Impact:** 3

**Recommendation:**

Weak Accept: I recommend accepting the paper, but will not argue for my recommendation if the majority of other reviewers have a different opinion.

**Summary:**

The paper focuses on the problem of last-mile navigation on image-goal navigation, which aims at reaching the goal after discovering it. It is also the key problem of many visual navigation methods but have barely been researched. The proposed method SLING is proved to raise the SOTA on success rate. Physical experiments are carried to prove the effectiveness on real environment.

**Issues:**

Add experiments on VGM.

**Quality Of The Limitations Section:**

Additional details required

**Reviewer Expertise:**

4: The reviewer is confident but not absolutely certain that the evaluation is correct

**Robotics Focus:**

Sufficient demonstration on hardware

**Strengths And Weaknesses:**

Strengths:
1)	The problem of last-mile navigation have barely been researched, despite it is proved to be the reason of many failures.
2)	This paper proposes a simple and interpretable module that can be used with many baselines. The experimental results are rich and impressive.
3)	Physical experiments are carried to demonstrate the effectiveness of transferring to real environment.

Weaknesses:
1)	There is no experiment that compares or combine SLING with VGM [1] which is the SOTA ImageNav method.
2)	It would be better to discuss some failure cases caused by adding SLING as a plug-in module, so it is more clear how SLING affects the baselines it is added to.

References:
[1] O. Kwon, N. Kim, Y. Choi, H. Yoo, J. Park, and S. Oh. Visual graph memory with unsupervised representation for visual navigation. In Proceedings of the IEEE/CVF International Conference on Computer Vision, pages 15890–15899, 2021.


**Summary Of Recommendation:**

The paper raises an interesting problem for deeper research, and proposes an effective method to tackle the problem with adequate experiments.

---

> ### Author Response · Authors · 2022-08-24
> **Response to Reviewer D7Jh**
>
> We thank D7Jh for their constructive feedback and appreciation for our research objective, simple-interpretable solution, and rich experiments including physical ones.
> > Compare or combine SLING with VGM [1] which is the SOTA ImageNav method
>
> Thanks for this suggestion. At the outset, let us gently clarify a nuanced point: ImageNav performance is highly correlated to the field-of-view (FoV). This is intuitive as an agent that sees more about the environment and associated context in one observation will do better. VGM [1] operates on panoramic observations and enjoys this advantage. However, other methods benchmarked in most prior works [3, 4, 5] and ours operated on a FoV of 90° (when comparing with NRNS [2], which used a FoV of 120°, we employ the same).
>
> For a fair comparison with VGM, we re-trained the best baseline from prior work, i.e. OVRL [3], now with panoramic images (instead of FoV of 90° as in the main paper). The basic implementation of panoramic OVRL (without any hp. tuning) performs better than VGM. Also, as suggested, we integrate SLING to this panoramic OVRL baseline and demonstrate sizable gains in both success rate and SPL. This shows that SLING (without customizations/tuning) works well in both limited FoV and panoramic image settings. Clear details are in the table below for the Gibson-curved split [2]:
>
> |Method                  |Overall Success ↑|Overall SPL ↑|
> |------------------------|---------------|-----------|
> |OVRL (90° FoV)          |45.6           |28.0         |
> |OVRL + SLING (90° FoV)  |__54.8__           |__37.3__       |
> |VGM (panoramic)         |74             |51         |
> |OVRL (panoramic)        |76.7          |59.4      |
> |OVRL + SLING (panoramic)|__78.6__          |__60.4__      |
>
> ***
> > It would be better to discuss some failure cases caused by adding SLING
>
> We gently point the reviewer to Sec. 4.5 where we discuss limitations, in an upfront manner. As requested, the concrete failure cases that arise due to including SLING are included below:
> 1. _Hyperlocal keypoints_: In a few cases, most of the keypoints are localized to a small area of the image. Here, SLING’s perspective-n-point estimates end up being of low quality. Usually, a simple solution would be to move the agent around, leading to new keypoint triggers. This in turn would pull the agent out of this local minima.
> 2. _Depth noise_: When depth observations have considerable noise, they affect policy predictions (Sec. 5.1 of [6]), including that of SLING. This was more prominent in the physical experiments we conducted across diverse scenes. Depth denoising schemes such as (1) cropping dead pixels from a label spectrogram followed by joint-bilateral filter to their neighbor pixels [7] and (2) layer depth denoising [8], have been integrated into existing toolkits [9,10] and show encouraging results for sim-to-real transfer [11]. These could be considered for future works.
> 3. _ImageNav’s disregard of goal orientation_: Since SLING finds overlap between agent and goal views, it must observe an image in a similar orientation to the goal to navigate the last mile. However, in its current form, success for the ImageNav task simply needs the agent to be within 1m from the goal image location (even if it’s facing the opposite direction to the goal image throughout the episode). SLING cannot exploit this nuance of the task definition. Having said that, we hold that the SLING approach is closely aligned to how humans attempt the ImageNav task.
>
> References:
> [1] O. Kwon et al. Visual graph memory with unsupervised representation for visual navigation. In CVPR, 2021.
> [2] M. Hahn et al. No rl, no simulation: Learning to navigate without navigating. In NeurIPS, 2021.
> [3] K. Yadav et al. Offline visual representation learning for embodied navigation. arXiv preprint arXiv:2204.13226, 2022.
> [4] Z. Al-Halah et al. Zero experience required: Plug & play modular transfer learning for semantic visual navigation. In CVPR, 2022.
> [5] E. Wijmans et al. Dd-ppo: Learning near-perfect pointgoal navigators from 2.5 billion frames. In ICLR, 2019.
> [6] Chattopadhyay et al. Robustnav: Towards benchmarking robustness in embodied navigation. In ICCV, 2021.
> [7] Camplani and Salgado. Efficient spatio-temporal hole filling strategy for kinect depth maps. In 3DIP, 2012.
> [8] Shen and Cheung. Layer depth denoising and completion for structured-light rgb-d cameras. In CVPR, 2013.
> [9] Grunnet-Jepsen and Tong. Depth post-processing for Intel® RealSense™ depth camera D400 series. https://dev.intelrealsense.com/docs/depth-post-processing (2020).
> [10] Fairo GitHub: [Hello Realsense](https://github.com/facebookresearch/fairo/blob/acea517e363d1f381cd5504a40c688d6cc9230ce/droidlet/lowlevel/hello_robot/remote/remote_hello_realsense.py#L106)
> [11] Yu et al. Visual-locomotion: Learning to walk on complex terrains with vision. In CoRL, 2021.

---

> ### Author Response · Authors · 2022-08-27
> **Rebuttal Summary and Updated Manuscript**
>
> From the review, we believe that the main questions were about:
>
> (1) Comparing SLING to VGM:
>
> Addressed by providing new results in panoramic observation space for a head-on comparison with VGM, which we improve over. We’ve integrated this study in Sec. 4.3 and the _new Appendix I_.
>
> (2) Include more failure cases:
>
> Beyond referencing our limitations section, we include additional details about failure cases, which we have also integrated into Sec. 4.5 and Appendix E.
>
> We believe that we concretely address all concerns shared in the review. Is our response and updated manuscript (reviewer-specific edits are in _blue_) satisfactory? If not, please let us know how we can better address these concerns.
>
> Thank you so much for your efforts to help improve our work!

---

### Comment · Area_Chair_m5Pb · 2022-08-19
**Meta Review Comments**

Thank you authors and reviewers, here is a short summary of some of the key strengths and weaknesses identified by the reviewers.
Authors & reviewers, please engage in a discussion regarding the issues raised by the individual reviews.

Strengths:
- the paper's focus on last mile navigation was perceived to be of high relevance and interest
- reviewers appreciated the clarity of writing and overall approach of the paper
- the presented experiments were considered of interest and worthwhile

Weaknesses:
- some reviewers requested additional experiments, e.g. to comparing/combining SLING and VGM, regarding the upper bound discussion (reviewer wKSA), etc.
- reviewers commented on potential difficulties in real world applications of the method, e.g. hyper parameter choices and potential brittleness
  of components of the approach

---

### Author Response · Authors · 2022-08-24
**General response and request for discussion**

In each response, we have conducted all the experiments the reviewers asked for. From methods operating on panoramic observations to last-mile specific baselines, from oracle last-mile navigation to increased tolerance to last-mile errors. All new experiments have reaffirmed that SLING is a plug-and-play addition that improves performance across IL, RL, and neural modular policies. This is also demonstrated by significant gains across diverse scene datasets in simulation as well as real-robot settings.

We thank the reviewers for their time and effort in helping improve our work. The experiment recommendations have definitely improved the impact of our research.

---

### Meta-Review · Area_Chair_m5Pb · 2022-09-05

**Recommendation:** Accept (Poster)
**Confidence:** 4

**Metareview:**

Concluding comments and observations:

The authors work introduces a navigation module structure and associated switches between goal discovery and "last mile navigation" for image-based navigation once a goal has been identified as being sufficiently close/prevalent in observed image sensor data.

The following are some of the key strengths and weaknesses in my view:

Strengths:
-  the last mile navigation phase vs exploration phase decomposition approach appears promising
-  the paper's core method is very simple to understand and implement and easy to combine with existing approaches
-  the paper and its revision provide an extensive experimental evaluation

Weaknesses:
-  the simplicity of the proposed method also means that there is little theoretical or deep methodological advance reported in this work - the focus is mostly on an apparently simple and effective approach and its empirical evaluation.

The reviewers were of similar opinions with 2 x weak accept and 1x reviewer who declared border line weak accept/weak reject in the discussion. I am recommending the work to be accepted.

I could appreciate how this paper stays true to a "simple method" and just focuses on the empirical evaluation thereof - I could see this method being appreciated and implemented in industrial / real world systems if it can be confirmed to increase performance as indicated. From a learning theory/deeper methodology contribution point of view it is however not so clear if there is a lasting contribution there.

**Best Paper Nomination:**

No